# The Effects of Ambient Temperature and Atmospheric Humidity on the Diffusion Dynamics of Hydrogen Fluoride Gas Leakage Based on the Computational Fluid Dynamics Method

**DOI:** 10.3390/toxics12030184

**Published:** 2024-02-28

**Authors:** Zhengqing Zhou, Yuzhe Liu, Huiling Jiang, Zhiming Bai, Lingxia Sun, Jia Liu, Wenwen Zhao

**Affiliations:** 1Research Institute of Macro-Safety Science, University of Science and Technology Beijing, Beijing 100083, China; 2School of Civil and Resource Engineering, University of Science and Technology Beijing, Beijing 100083, China; 3Kunlun Digital Technology Co., Ltd., Beijing 100007, China

**Keywords:** hydrogen fluoride (HF), leakage dispersion, ambient temperature, atmospheric humidity, Computational Fluid Dynamics (CFD)

## Abstract

In order to investigate the impact of environmental temperature and atmospheric humidity on the leakage and diffusion of hydrogen fluoride (HF) gas, this study focused on the real scenario of an HF chemical industrial park. Based on the actual dispersion scenario of HF gas, a proportionally scaled-down experimental platform for HF gas leakage was established to validate the accuracy and feasibility of numerical simulations under complex conditions. Using the validated model, the study calculated the complex scenarios of HF leakage and diffusion within the temperature range of 293 K to 313 K and the humidity range of 0% to 100%. The simulation results indicated that different environmental temperatures had a relatively small impact on the hazardous areas (the lethal area, severe injury area, light injury area, and maximum allowable concentration (MAC) area) formed by HF gas leakage. At 600 s of dispersion, the fluctuation range of hazardous area sizes under different temperature conditions was between 3.11% and 13.07%. In contrast to environmental temperature, atmospheric relative humidity had a more significant impact on the dispersion trend of HF leakage. Different relative humidity levels mainly affected the areas of the lethal zone, light injury zone, and MAC zone. When HF continued to leak and disperse for 600 s, compared to 0% relative humidity, 100% relative humidity reduced the lethal area by 35.7%, while increasing the light injury area and MAC area by 27.26% and 111.6%, respectively. The impact on the severe injury area was relatively small, decreasing by 1.68%. The results of this study are crucial for understanding the dispersion patterns of HF gas under different temperature and humidity conditions.

## 1. Introduction

HF is a typical toxic and harmful heavy gas widely used in various industrial processes. If released into the atmosphere, it can cause severe harm to human health, the environment, and infrastructure. Even after natural attenuation, HF can still have negative effects on plant growth once it comes into contact with soil [1]. The long-term inhalation of HF can lead to serious consequences, such as systemic toxicity, cardiac arrest, and death, and can have permanent effects on human health [2,3,4,5]. One characteristic of the consequences of HF leakage accidents is the depth of the harm. From 2016 to 2020, there were a total of six fatal accidents due to HF leakage in chemical workshop enterprises in Weifang, Shandong, Rugao, Jiangsu, Ganzhou, Jiangxi, Guiyang, Guizhou, and Nantong, Jiangsu, resulting in 12 fatalities. In January 2016, there was an HF leakage accident at the Tetrafluorobenzyl alcohol workshop of Shandong Weifang Changxing Chemical Co., Ltd. (Weifang, China), resulting in three deaths and one injury. In December 2018, there was an HF leakage accident at the Fluoropyrimidine synthesis workshop of Zhongchang Chemical Co., Ltd. (Rugao, China), resulting in three deaths and one injury. In January 2020, there was an HF leakage accident at Jiangxi Shilei Fluorine Chemical Co., Ltd. (Ganzhou, China), resulting in three deaths and one injury. In August 2020, there was an HF leakage accident at Guiyang Wengfu Kaiphos Fluorosilicon New Materials Co., Ltd. (Guiyang, China) resulting in two deaths. In December 2020, there was an anhydrous HF leakage accident at the Lithium Hexafluorophosphate production workshop of Houcheng Technology (Nantong) Co., Ltd. (Nantong, China), resulting in one death [6]. Another characteristic of the aftermath is the wide-ranging impact. In 2009, a significant HF leakage occurred in a chemical enterprise in the United States, leading to the emergency evacuation of nearly 10,000 people in the surrounding area. In 2012, a similar incident occurred in a chemical enterprise in South Korea [7], where the leaked HF accumulated for over 10 days, causing vegetation damage within a radius of 1 km. In 2014, a port in Taiwan experienced an HF leakage incident, prompting the urgent evacuation of people within approximately 100 m of the leakage source. Therefore, understanding the leakage and dispersion of hydrogen fluoride gas is crucial for assessing and mitigating potential hazards.

The large-scale full-size on-site experimental method is the most direct approach for studying the diffusion of toxic gases [8,9,10,11,12]. However, considering the characteristics and hazards of toxic gases, as well as the human and material resources required to establish an experimental platform, large-scale on-site experimental methods have certain limitations. Therefore, since the 1980s, wind tunnel experiments have been widely used internationally for the study of gas leakage and diffusion [13,14,15,16]. Wind tunnel experiments, compared to large-scale full-size on-site experiments, reduce costs and difficulties. However, monitoring gas concentrations in wind tunnel experiments still requires a substantial investment. Thus, with the development of computer technology, low-cost and safe Computational Fluid Dynamics (CFD) methods have become one of the mainstream approaches for studying gas leakage and dispersion. Yang [7] adopted CFD to simulate the accident and compared it with post-incident field investigation data. The study found that the contour lines of equivalent toxic doses calculated through a simulation (based on a height of 1.5 m) compared with the distribution points of toxic plants collected after the incident. It was found that the estimated damage area boundaries matched the simulation results consistently, particularly downwind to the east of the incident site. Tauseef [17] et al. employed the finite volume method to solve the density regulation equation group, using a simplified approach to solve the pressure–velocity coupling, and employed the realizable k-ε turbulence model to simulate the diffusion of dense gas in the presence of obstacles. They argued that the realizable k-ε model is more suitable for the CFD simulation of dense gas diffusion in the presence of obstacles compared to other turbulence models. Fiates [18] et al. conducted a study on the diffusion of heavy gases, such as CO_2_ and LNG, using the open-source CFD code OpenFOAM. The results showed that the modified OpenFOAM could handle the diffusion of heavy gases well, with simulation results consistent with experiments and other software, validating the application prospects of open-source CFD software in this regard. Yoshie [19] et al. studied the diffusion of gas and heat behind a high-rise building under unstable non-isothermal turbulent boundary layer conditions using wind tunnel experiments and CFD simulation techniques. The results showed that the inflow turbulence has a significant impact on the turbulent structure of large eddy simulation (LES), and correctly simulating the inflow turbulence conditions is crucial for LES calculations at the inlet boundary. Ohba [20] et al. investigated the characteristics of LNG leak cloud dispersion using wind tunnel experiments and numerical simulation methods. The results showed that the concentration mean and standard deviation calculated using the direct numerical simulation (DNS) method also matched the wind tunnel results. Wind tunnel testing and numerical simulation can effectively simulate the dispersion patterns of LNG leak clouds under different atmospheric stability conditions. Hanna [21] et al. used FLACS to simulate two significant chlorine gas release incidents. The results showed that buildings can affect the speed and extent of dense cloud movement. Additionally, they compared FLACS simulation results with several widely used simplified gas dispersion model results, showing that the concentration predicted by FLACS roughly falls within the range of predictions by other models, indicating FLACS’ ability to provide an additional analysis of detailed architectural influences. Dadashzadeh [22] et al. studied the distribution of the combustion product emissions CO, NO_2_, and CH_4_ in different areas of equipment and the risk of personnel inhaling toxins during a liquefied natural gas leakage fire accident on an offshore drilling platform using FLACS and risk assessment methods. They believed that considering the exposure time in risk assessment methods is more reasonable in assessing the toxicity risk of emission mixtures compared to concentration methods. Souza [23] et al. used CFD simulation combined with Latin hypercube sampling statistical techniques to study the leakage and diffusion conditions of flammable gases under various circumstances. Based on CFD results, they proposed an empirical relationship. Compared with existing formula predictions, the results predicted by this new formula are on average 117% smaller, allowing for a more accurate calculation of the range and volume of hazardous areas and reducing the cost of industrial safety devices.

Liang [24] et al. believed that CFD technology has matured and can be widely applied to the study of urban micro-meteorological processes. However, they also pointed out challenges, such as the optimization of ultra-fine grids, improving the integration of different scale models, addressing data limitations, and simulating more climate conditions that need further resolution. Previous research on the dynamic response process of the diffusion of toxic and harmful gases has primarily focused on the influence of factors, such as the wind speed, wind direction, and leakage rate, on gas diffusion trends. Galeev [25] used Fluent software to simulate the diffusion of liquid nitrogen, finding that gravity led to the formation of a toxic zone in the upwind direction. When studying the influence of wind, the impact area was largest at a wind speed of 1 m/s. Paloma [26], based on CFD calculations, discussed the influence of wind direction and wind speed on the volume of gas clouds. The results showed that the size of the gas plume varied with changes in wind and depended on the gas concentration. Concentrated gas releases upwind promoted plume dilution, increased the degree of harm, and simultaneously reduced the volume of the plume. Shao [27] studied the diffusion patterns under the influence of temperature, wind speed, and wind direction when chlorine gas leaked from a chemical laboratory and its surrounding urban area in Beijing using numerical simulation methods. The results revealed that the direction of chlorine gas dispersion was influenced by wind direction, and the range of dispersion was affected by temperature and wind speed. The high-temperature exposure area at a high risk was 21.05% larger than the high-risk area at low temperatures. When the wind direction was opposite to the direction of the buildings, the exposed area of high risk was 78.95% smaller than that in the direction of the buildings. Yue [28], using STAR-CCM+ software and numerical simulation, investigated the influence of wind speed on gas dispersion patterns and the concentration distribution.

Among various environmental factors, ambient temperature and atmospheric humidity play a pivotal role in influencing the dispersion characteristics of HF gas. Temperature changes impact the kinetic energy and velocity of gas molecules, thereby affecting the density and dispersion pattern of HF. Additionally, humidity levels influence gas density by introducing water molecules into the gas mixture, further altering its dispersal characteristics. Despite extensive research on the leakage and dispersion of HF by previous scholars, there has been limited investigation into the specific impact of ambient temperature and atmospheric humidity on the leakage and dispersion of high-frequency gases. Therefore, there is an urgent need for a comprehensive study on the influence of temperature and humidity on the leakage and dispersion of high-frequency gases to deepen our understanding of related risks.

This study utilizes CFD to investigate the influence of environmental temperature ranging from 293 K to 313 K and atmospheric humidity ranging from 0% to 100% on the release and dispersion of HF gas. Taking the complex scenario of a Hydrofluoric Acid production line in Zhejiang Quzhou Industrial Park as a prototype, a scaled gas leakage and dispersion experimental platform is constructed with a geometric ratio of 1:100. In the experiments, the relatively safe CO_2_ gas is used to simulate the toxic HF gas in the leakage and dispersion process. Experimental gas concentration values are compared with numerical simulation results to validate the reliability of the numerical simulation method in studying gas leakage and dispersion in complex scenarios. By gaining in-depth insights into the impact of temperature and humidity on HF dispersion, we can optimize safety protocols, design more efficient sealing systems, and formulate specific emergency measures tailored to particular environmental conditions.

## 2. Methodology

### 2.1. Governing Equation of CFD Model

Ansys Fluent is a solver based on the finite volume method, addressing equations in the Navier–Stokes form, particularly the momentum conservation equation, as depicted in Equation (1). The convergence of the solution is assessed by ensuring mass, momentum, and energy conservation in each control volume (grid) across the computational domain. The equations governing mass and energy conservation [29] are presented in Equations (2) and (3), respectively. The diffusion process of HF is expected to satisfy the following control equations.
(1)$$\frac{\partial }{\partial t}\left(\rho \vec{u}\right)+\nabla \cdot \left(\rho \vec{u}\vec{u}\right)=-\nabla p+\nabla \cdot \left(\overset{=}{\tau }\right)+\rho \vec{g}$$
(2)$$\frac{\partial \rho }{\partial t}+\nabla \cdot \left(\rho \vec{u}\right)=0$$
(3)$$\frac{\partial \left(\rho c_{v}T\right)}{\partial t}+\nabla \cdot \left(\rho \vec{u}c_{p}T\right)=\nabla \cdot \left(k_{T}\nabla T\right)$$
where *ρ* represents the density of the fluid, kg/m^3^, while *u* denotes the velocity of the particle, m/s. *p* signifies absolute pressure, Pa. The turbulent thermal conductivity is denoted by *k_T_*, W/(m·K). *τ* represents shear stress, N/m^2^, and *c_v_* and *c_p_*, respectively, represent the specific heat of the fluid, J/(kg·K).

The k-ε series models primarily consist of three types: the Standard k-ε, RNG k-ε, and Realizable k-ε models. The Standard k-ε model is proposed based on extensive operational experience and experimental phenomena, neglecting the effects of fluid molecular viscosity, making it suitable for fully turbulent scenarios. The RNG k-ε model, based on statistical techniques, improves the ε equation and supplements the effects of viscosity in low-Reynolds-number flows, providing higher accuracy. In comparison to other models in the k-ε series, the Realizable k-ε model [30] retains the realistic conditions of fluid anisotropy during turbulent flow. It deeply considers situations where fluid rotates on curved surfaces, and in computing the turbulent viscosity coefficient, it includes variables related to rotation and curvature. This enhances the accuracy of solving fluid flow around cylindrical tank geometries. Therefore, in this study, the Realizable k-ε model is selected as the turbulence model, and its equations are shown in Equations (4) and (5).
(4)$$\frac{\partial \left(\rho k\right)}{\partial t}+\frac{\partial \left(\rho ku_{i}\right)}{\partial x_{i}}=\frac{\partial }{\partial x_{i}}\left[\left(\mu +\frac{\mu_{t}}{\sigma_{k}}\right)\cdot \frac{\partial k}{\partial x_{j}}\right]+G_{k}-\rho \varepsilon$$
(5)$$\frac{\partial \left(\rho \varepsilon \right)}{\partial t}+\frac{\partial \left(\rho \varepsilon u_{i}\right)}{\partial x_{i}}=\frac{\partial }{\partial x_{i}}\left[\left(\mu +\frac{\mu_{t}}{\sigma_{\varepsilon }}\right)\cdot \frac{\partial \varepsilon }{\partial x_{j}}\right]+\rho C_{1}E\varepsilon -\rho C_{2}\frac{\varepsilon^{2}}{k+\sqrt{\nu \varepsilon }}$$
where *k* represents turbulent kinetic energy in joules, J, and *ε* represents the dissipation rate of turbulent kinetic energy, *σ_k_* = 1.0, *σ_ε_* =1.2, *C*_1_ = max$\left(0.43,\frac{\eta }{\eta +5}\right)$, *C*_2_ = 1.92, $\eta =\frac{k}{\varepsilon }\sqrt{2E_{ij}\cdot E_{ij}}$, $E_{ij}=\frac{1}{2}\left(\frac{\partial u_{i}}{\partial x_{j}}+\frac{\partial u_{j}}{\partial x_{i}}\right)$, *G_k_* represents the shear force term induced by the average velocity gradient, and this term can be expressed by Equation (6).
(6)$$\begin{array}{l} G_{k}=2\mu_{t}\left[{\left(\frac{\partial u}{\partial x}\right)}^{2}+{\left(\frac{\partial v}{\partial y}\right)}^{2}+{\left(\frac{\partial w}{\partial z}\right)}^{2}\right]\\ +\mu_{t}{\left(\frac{\partial u}{\partial x}+\frac{\partial v}{\partial y}\right)}^{2}+\mu_{t}{\left(\frac{\partial u}{\partial x}+\frac{\partial w}{\partial z}\right)}^{2}+\mu_{t}{\left(\frac{\partial v}{\partial y}+\frac{\partial w}{\partial z}\right)}^{2} \end{array}$$
where *μ_t_* represents the turbulent viscosity coefficient, which can be calculated using Equations (7) and (8).
(7)$$\mu_{t}=\frac{C_{\mu }\rho k^{2}}{\varepsilon }$$
(8)$$C_{\mu }=\frac{1}{A_{0}+A_{s}U^{*}k/\varepsilon }$$
where *A*_0_ is a constant, *A_s_* is a parameter related to the fluid angular velocity in the flow field, and *U** is the time-averaged rotation rate tensor. This term considers fluid rotational motion, which is one of the key features distinguishing the Realizable k-ε model from other models in its series.

### 2.2. Modeling and Verification

To validate the reliability of the CFD simulation method, this study employed a scaled-down experimental model. The Fluent 6.3 software (Ansys, Canonsburg, PA, USA) was utilized to accurately reconstruct the real-world scenario, forming the basis of subsequent simulations of gas dispersion.

#### 2.2.1. Scaling Experiment

This study selected a specific area of the HF production line in a typical fluorine chemical enterprise in Quzhou, Zhejiang (see Figure 1a), for modeling. The research focused on the diffusion of HF gas leakage in the production line area, and a three-dimensional model was established based on the real scene (see Figure 1c). Utilizing similarity theory [4], a scaled experimental model was constructed, and the accuracy of the Computational Fluid Dynamics (CFD) simulation was verified based on experimental data from the scaled model. Considering the toxicity of HF gas, CO_2_ gas with a concentration close to HF was chosen as a substitute gas in the scaled experiments. This was performed to validate the selected control equations, turbulence models, three-dimensional models, and the grid division, demonstrating their applicability in calculating the diffusion process of toxic and harmful heavy gases in complex scenarios in this study. The scaled experimental model adopts a 1:100 geometric similarity ratio to ensure that the linear dimensions of the model correspond proportionally to the real scenario. For instance, if the actual diameter and height of the leaking tank are 16 m and 15 m, respectively, the corresponding model dimensions are 16/100 m and 15/100 m. Ultimately, a successfully proportionally scaled-down model is established, as illustrated in Figure 1b.

In designing experimental conditions, factors, such as the leakage rate and environmental wind speed, were taken into consideration. To precisely control the leakage rate, flowmeters and pressure-reducing valves were employed, establishing a leakage rate experimental group with conditions of 2.5, 7.5, and 12.5 L/min. For controlling environmental wind speed, adjustments were made by varying fan speeds and using a handheld anemometer (SMART SENSOR, Dongguan, China), setting four conditions corresponding to wind levels 0, 1, 2, and 3. Conversion based on the similarity theory allowed for the translation of real-world wind levels into wind speeds in the scaled-down model, as detailed in Table 1.

The leaked CO_2_ gas itself is colorless and odorless, making it challenging to observe experimental phenomena with the naked eye. Therefore, this study quantifies the diffusion characteristics of the gas by analyzing the numerical changes in the concentration at monitoring points.

To monitor the distribution of and variations in CO_2_ gas concentrations in the fluid domain above the scaled-down model, nine monitoring points were uniformly distributed on a sandbox. CO_2_ gas concentration sensors were fixed at each monitoring point, as illustrated in Figure 2. Notably, Position 1 serves as the location of the leak, with the other monitoring points numbered in ascending order based on their distances from the leak point. Specifically, Points 2 and 3 form the closest group, followed by Point 4, and subsequently, Points 5 and 6 and Points 7 and 8, with Point 9 being the farthest. The overall structure of the experimental platform is depicted in Figure 3.

#### 2.2.2. Model Validation

Partial physicochemical parameters of HF and carbon dioxide are shown in Table 2. During the model verification process, relevant parameters were inputted to ensure complete consistency with the boundary conditions of the scaled experimental model scenario, including identical boundary conditions and environmental parameters. To validate the accuracy and reliability of the numerical simulation, a geometric model consistent with the real scenario was constructed. Simultaneously, to ensure computational efficiency, the model underwent appropriate simplifications. The concentration data obtained from numerical calculations were compared with the data measured in the scaled experiments.

Due to the complexity of the research scenario, further reliability validation of the numerical model is necessary. In this study, a complex scenario of gas leakage similar to the experimental platform was constructed, and multiple experiments were conducted. A three-dimensional geometric model identical to the complex real scenario was constructed. Simultaneously, the same boundary conditions and environmental parameters as the experimental group were set for simulation reconstruction, comparing the concentration data obtained from numerical calculations with experimental data. Different leakage rates were set in the scaled-down experimental design, with the gas concentration data at monitoring Point 4 selected as the validation dataset. The comparison between experimental and simulated values is shown in Figure 4.

From Figure 4, it can be observed that the concentration values measured at monitoring Point 4 under three different leakage rates exhibit a consistent increasing trend through both measurement methods. However, due to the relatively large volume of the CO_2_ gas concentration sensor, which acts as an obstacle in the fluid domain, it influences the diffusion path and concentration distribution of the gas, causing CO_2_ to accumulate in the vicinity. Consequently, experimental values tend to be higher than simulated values. Nevertheless, it is noteworthy that the difference between experimental and simulated values in each comparison group is not substantial. Relative error calculations for the three datasets yield average relative deviations of 13.54%, 15.94%, and 18.02% at leakage rates of 2.5, 7.5, and 12.5 L/min, respectively. This level of deviation falls within an acceptable range, indicating the reliability of the numerical model used in this study. It is suitable for simulating the leakage and diffusion of toxic and hazardous gases in complex scenarios. Combined with the physical parameters of HF validated in the previous work of our research group [31], it can be used to numerically simulate the process of HF gas leakage and diffusion.

## 3. Simulation Application Background

### 3.1. Classification of Hazardous Areas

For varying concentrations of HF, the human body exhibits distinct physiological responses. As per the stipulations in GBZ2.1-2019, the highest allowable concentration for HF is set at 2 mg/m^3^. Considering both national standards and the severity of HF’s impact on the human body, four danger zones have been delineated: the lethal area, severe injury area, light injury area, and the maximum allowable concentration area (the MAC area) [31].

### 3.2. Actual Environmental Conditions and Boundary Conditions

In the design of the simulation scenario, it is imperative to ensure that all simulated conditions, including environmental wind direction, wind speed, temperature, and humidity, strictly adhere to the actual conditions of the real scenario. Detailed temperature and humidity data from Quzhou City, Zhejiang Province, for the years 2019 to 2022, have been collected. The results indicate that during the years 2019 to 2022, the temperature fluctuated mainly within the range of −5 °C to 40 °C, and the relative humidity varied between 0% and 100%. Given that the gas selected for this study is HF, we consider only the impact of temperature on dispersion outcomes when HF is in a gaseous state. Considering HF’s boiling point of 19.51 °C, we chose to simulate temperatures within the range of 19.51 °C to 40 °C, with a simulated relative humidity range of 0% to 100%.

In this study, the storage pressure of the HF tank is 0.3 MPa, and the leakage source is a circular hole with a radius of 0.01 m. In the scenario considered in this study, the release of HF gas corresponds to critical flow. The calculated mass flow rate of HF gas leakage is 0.18 kg/s (see Table 3).

## 4. Results and Discussion

### 4.1. Patterns of Change in Hazardous Areas at Different Ambient Temperatures

To investigate the impact of ambient temperature (T) on the HF concentration distribution in complex scenarios, this study simulated conditions in the range of 293–313 K for this scenario. For a clearer comparison, we examined the three-dimensional concentration distribution of HF under conditions of 293 K and 313 K, as shown in Figure 5. From the figure, it can be observed that at different ambient temperatures, HF primarily disperses in the direction of the leak after the leakage, gradually spreading in all directions over time, forming a block-shaped affected area. Observing the block-shaped affected area reveals that regions closer to the leak and the bottom of buildings have higher HF concentrations. In areas with dense obstacles, leaked HF accumulates, while in areas with sparse or no obstacles, HF disperses more rapidly and is more intensely diluted by ambient air. This indicates that the distribution of objects in the environment, such as buildings, influences the diffusion behavior of HF following a leakage incident, with the presence of obstacles restricting the gas’s diffusion range. Conversely, in open areas, HF disperses more rapidly and undergoes greater dilution by the surrounding air. Comparing the three-dimensional concentration distributions at the same moment for T at 293 K and 313 K, overall, the diffusion area slightly increases with the temperature rise. This phenomenon may be attributed to the elevation in temperature leading to an increase in gas molecule velocity, thereby promoting the diffusion and mixing processes of HF molecules. Therefore, the influence of temperature on the diffusion range of HF is significant and should not be overlooked.

To visually inspect the areas of various danger zones more effectively, we selected two-dimensional planes at a height of 1.5 m for two conditions (T at 293 K and 313 K). Using the MAC area, the severe injury area, the light injury area, and the lethal area four concentration levels as boundaries, a color legend was set with red, yellow, green, and light blue representing the lethal, the severe injury, the light injury, and the MAC areas, respectively. The corresponding distribution of danger zones is shown in Figure 6. From the figure, it can be observed that under different ambient temperatures, the distribution of HF danger zones remains consistent, with the lethal, the severe injury, the light injury, and the MAC areas appearing from the innermost to outermost areas. Except for the lethal area, which is block-shaped, other areas are distributed in a surrounding manner. Upon closer observation of the danger zones, it is evident that in areas with concentrated obstacles, the expansion of danger zones is hindered. In areas with sparse or no obstacles, the expansion of danger zones is faster, and the bandwidths of the severe injury, the light injury, and the MAC areas significantly increase. Additionally, compared to 293 K, when T is 313 K, the differences in the block-shaped impact area and the distribution characteristics of danger zones formed by HF gas leakage and diffusion are relatively small, with the impact area slightly larger under higher-temperature conditions. This is because as the ambient temperature increases, the kinetic energy of HF molecules increases, intensifying their random motion.

To further investigate the impact of environmental temperature on danger zones, we quantified the areas of four danger zones at a height of 1.5 m under conditions where T ranged from 293 K to 313 K. The results are presented in Figure 7.

Based on the results in Figure 7, within the continuous 600 s of HF leakage, the areas of lethal and severe injury zones steadily increase, while the areas of light injury and MAC zones exhibit fluctuations but overall show an increasing trend. It is noteworthy that the growth rates of the severe injury, the light injury, and the MAC zones are gradually rising. The areas of different zones are arranged in descending order: lethal, severe injury, MAC, and light injury. This indicates that once HF leaks and spreads, the damage to individuals will be quite severe, even irreversible. Furthermore, if the leakage source is not promptly addressed, it will result in extensive danger zones, leading to serious consequences for the enterprise area and the surrounding environment. Simulation results demonstrate that within the temperature range of 293 K to 313 K, the influence of different environmental temperatures on the hazardous zones formed by HF gas leakage and diffusion is relatively small. After 600 s of leakage, the fluctuation range of hazardous zones caused by different temperatures ranges from 3.11% to 13.07%. The overall areas of these hazardous zones are proportional to the environmental temperature. At 600 s of leakage, the impact of different environmental temperatures on the severe injury and the MAC areas is relatively significant. Compared to 293 K, when the temperature is 313 K, the area of the severe zone increases by 10.24%, and the MAC area increases by 13.07%. Conversely, the impact on the lethal and light injury areas is relatively small. Compared to 293 K, when the temperature is 313 K, the area of the lethal zone increases by 3.11%, and the area of the light injury zone increases by 4.21%. The overall increase in the hazardous zone is 6.22% (see Table 4 for details).

### 4.2. Patterns of Change in Hazardous Areas at Different Ambient Relative Humidities

To thoroughly investigate the impact of relative humidity on the concentration distribution of HF leakage in complex scenarios, we conducted meticulous simulation calculations for various conditions within the range of relative humidity (*φ*) from 0% to 100%. To highlight the contrasting effects, we selected the 3D concentration distribution of HF under extreme environmental relative humidity conditions of 0% and 100%, as illustrated in Figure 8. The figure reveals that the three-dimensional concentration distribution characteristics of HF and the morphology of the affected zones are consistent across different atmospheric humidities. Similarly, under different atmospheric relative humidity conditions, HF primarily disperses in the direction of the leak after release, gradually spreading in all directions over time, forming block-shaped affected areas. Regions near the leakage point and the bottom of buildings exhibit higher HF concentrations. In areas densely populated with obstacles, leaked HF tends to accumulate, while in regions with sparse or no obstacles, HF disperses more rapidly. A comparative analysis of results under different relative humidity conditions indicates that during the early stages of leakage (0–300 s), the impact of relative humidity on the formation of block-shaped affected zones by HF is not prominent. However, during the later stages of leakage (300–600 s), a higher relative humidity results in a smaller range of block-shaped affected zones, leading to a significant overall reduction in the concentration. This phenomenon is attributed to the presence of water vapor in the air, which alters the proportional composition of various gas components and influences the diffusion of HF. As the relative humidity increases, the air density decreases, intensifying the diffusion movement of HF. Consequently, in complex building environments, the “clustering” effect of HF gas becomes less pronounced, facilitating the dispersion of HF gas.

To visually assess the areas of various hazardous zones, we selected two-dimensional planes at a height of 1.5 m under two conditions of relative humidity (*φ*)—0% and 100%. The corresponding distribution of hazardous zones is depicted in Figure 9. Through the observation of the figure, it is evident that the influence patterns and expansion characteristics of relative humidity on the distribution of HF hazardous areas remain consistent with the effects of temperature. Within the range of 0% to 100% relative humidity, the interaction between water molecules and HF molecules significantly affects the diffusion behavior of HF in the air. Under 0% relative humidity conditions, the scarcity of water molecules in the air makes HF molecules more prone to diffusion, resulting in the formation of larger hazard zones in the surrounding environment. However, under 100% relative humidity conditions, the increase in water molecules in the air leads to chemical reactions with HF molecules, forming products, such as hydrofluoric acid, thereby inhibiting the diffusion of HF and limiting the expansion of hazard zones. Additionally, environmental factors, such as buildings and obstacles, also influence the size and shape of the hazard zone. Under high-humidity conditions, the surfaces of buildings may become damp due to the presence of water vapor, altering the adsorption and diffusion behavior of HF molecules on building surfaces. This may cause changes in the shape of the hazard zone and partially restrict its expansion. Comparing different relative humidity conditions, we observed that in the later stages of the leakage, as the relative humidity increases, the block-shaped lethal areas become smaller. The changes in severe injury areas are not significant, while the bandwidths of the light injury and the MAC areas notably increase. In other words, during the later stages of the leakage, the expansion of lethal areas is constrained to some extent with the increase in atmospheric humidity. Under the premise of minimal changes in the total hazardous area, the expansion of the light injury and the MAC areas becomes more pronounced.

To further investigate the influence of relative humidity on hazardous areas, we quantitatively analyzed the areas of four hazardous zones at a height of 1.5 m under relative humidity (*φ*) ranging from 0% to 100%, as depicted in Figure 10. As observed in the figure, within 600 s of continuous leakage and dispersion of HF, the areas of the four hazardous zones consistently increased with varying relative humidity. The sequence of their sizes is as follows: the lethal area consistently maintained the largest area, while the light injury area consistently had the smallest area. Notably, in the later stages of the leakage, higher atmospheric humidity led to a larger MAC area, even surpassing the severe injury area. Conversely, lower atmospheric humidity resulted in a smaller MAC area, falling below the severe injury area. The area of the lethal area exhibited an inverse proportionality to relative humidity (see Figure 10a). At any given moment, the growth rate of the lethal area was also inversely proportional to relative humidity. When the atmospheric relative humidity reached 100%, the growth rate of the lethal area gradually decreased in the later stages of the leakage until it approached zero. The MAC area showed a direct proportionality to relative humidity (see Figure 10d). For relative humidity in the range of 0% to 40%, the MAC area continued to increase, but the rate of increase was relatively small. In the later stages, fluctuations occurred, and there were instances of area reduction. For relative humidity in the range of 60% to 100%, the growth rate of the MAC area increased significantly with the rise in atmospheric humidity. As depicted in Figure 10b, the influence of relative humidity on the severe injury area was relatively minimal. In Figure 10c, it can be observed that the impact of relative humidity on the light injury area was minor in the early stages of the leakage. In the later stages, relative humidity induced changes in area, but without a specific pattern.

As depicted in Figure 10, the influence of different humidity levels on the lethal, the light injury, and the MAC areas is pronounced within the range of 0% to 100% relative humidity. During the continuous leakage and dispersion of HF, the increase in atmospheric humidity inversely affects the area of the lethal zone. When the diffusion reaches 600 s, the lethal area decreases by 35.7% when the relative humidity is 100% compared to 0%. In contrast, the areas of the light injury and the MAC areas show a direct proportionality with atmospheric humidity. At 100% relative humidity, compared to 0%, the MAC area increases by 111.6%, and the light injury area increases by 27.26%. The impact on the severe injury area is relatively minor, with a decrease of 1.68% when the relative humidity is 100% compared to 0%. It is evident that the increase in atmospheric humidity, under the condition of minimal influence on the overall dispersion area (refer to Table 5), significantly inhibits the expansion of the lethal area.

## 5. Conclusions

This study employs the CFD simulation method to systematically investigate the influence of environmental temperature and atmospheric humidity on the dynamic response of HF gas dispersion after a leak. Additionally, a scaled-down experimental model, constructed at a 1:100 ratio, is used for model validation. The results reveal the impact of environmental temperature and atmospheric humidity on the hazardous zones formed by HF leakage, leading to the following conclusions:

(1) Within the temperature range of 293–313 K, the influence of different temperatures on the dispersion of HF gas leakage and the formation of hazardous areas is relatively limited. Under a 600 s leakage scenario, the fluctuation range of hazardous areas caused by different temperatures is between 3.11% and 13.07%. During this period, the impact of different ambient temperatures on areas of severe injury and the MAC areas is relatively significant. Compared to 293 K, at 313 K, the area of severe injury increases by 10.24%, and the MAC area increases by 13.07%. Conversely, the impact on lethal and light injury areas is relatively small. Compared to 293 K, at 313 K, the lethal area increases by 3.11%, and the light injury area increases by 4.21%. The overall hazardous area increases by 6.22%.

(2) When the atmospheric relative humidity is within the range of 0–100%, the influence of different humidity levels on the lethal, the light injury, and the MAC areas is relatively more pronounced compared to the impact of ambient temperature. During the dynamic process of continuous HF leakage and dispersion, the increase in atmospheric humidity is inversely proportional to the area of the lethal area. When the diffusion reaches 600 s, at 100% relative humidity, the lethal area decreases by 35.7% compared to that with 0% humidity. The areas of the light injury and the MAC areas are directly proportional to the atmospheric relative humidity. At 100% relative humidity, compared to 0%, the MAC area increases by 111.6%, and the light injury area increases by 27.26%. With 0% relative humidity, the impact on the area of severe injury is relatively small, with a decrease of 1.68%. Overall, the increase in atmospheric humidity significantly inhibits the expansion of the lethal areas, although it has a minor impact on the overall hazardous area.

## Figures and Tables

**Figure 1 toxics-12-00184-f001:**
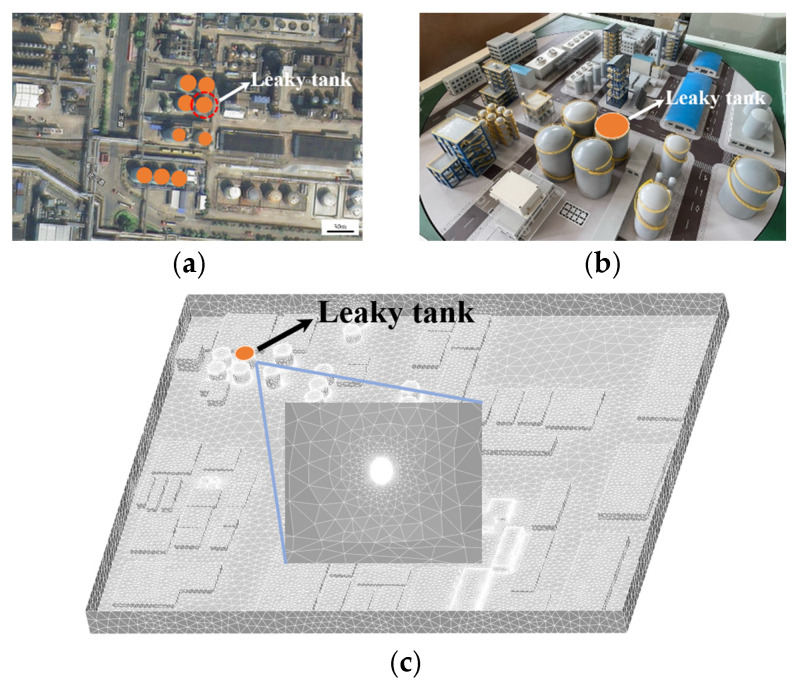
Model construction, (**a**) Complex scene satellite map image; (**b**) Scaled-down model; (**c**) 3D model.

**Figure 2 toxics-12-00184-f002:**
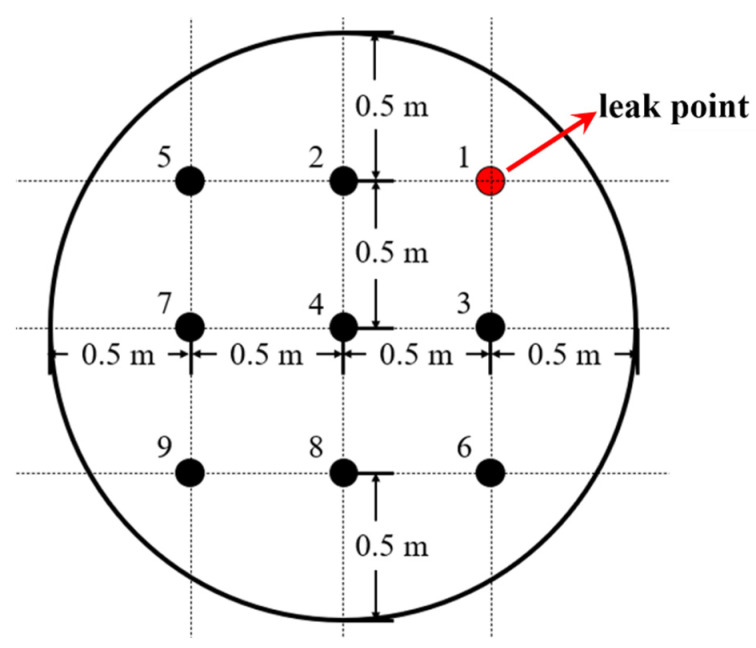
CO_2_ concentration sensor deployment.

**Figure 3 toxics-12-00184-f003:**
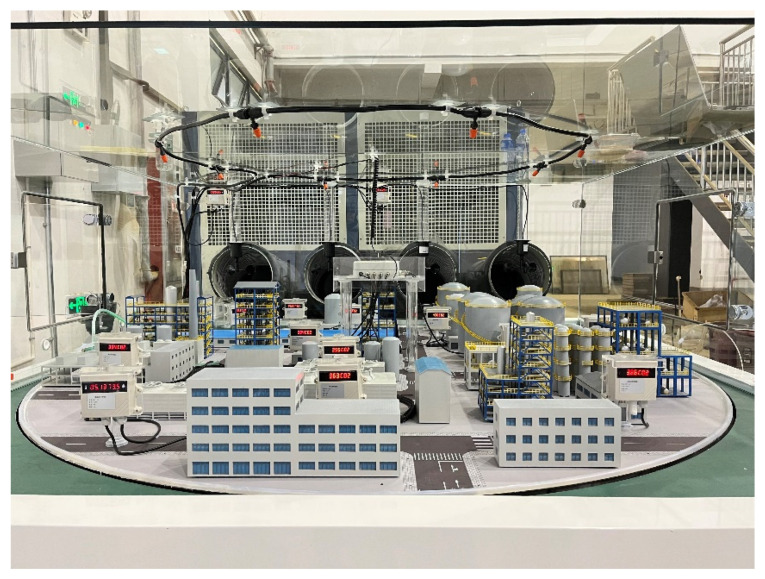
Overall structure of the experimental platform.

**Figure 4 toxics-12-00184-f004:**
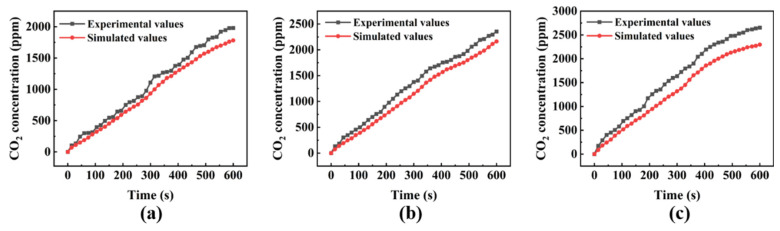
Comparison of experimental and simulated values for different leakage velocities at monitoring Point 4, (**a**) 2.5 L/min; (**b**) 7.5 L/min; (**c**) 12.5 L/min.

**Figure 5 toxics-12-00184-f005:**
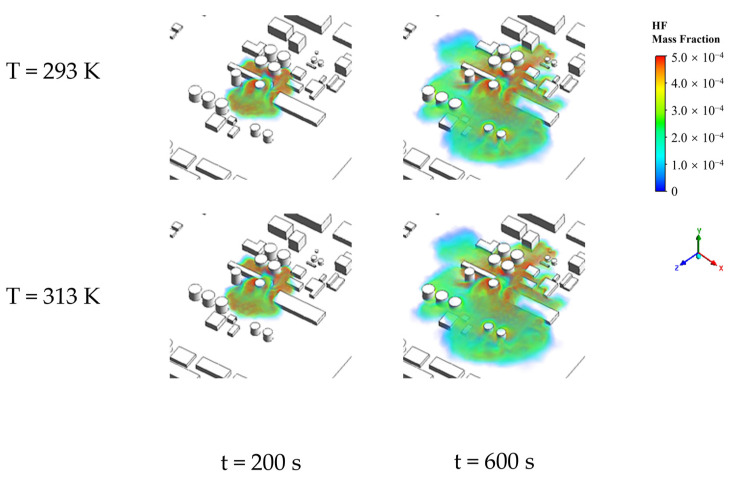
Three-dimensional concentration distribution of HF for complex scenes (T = 293, 313 K).

**Figure 6 toxics-12-00184-f006:**
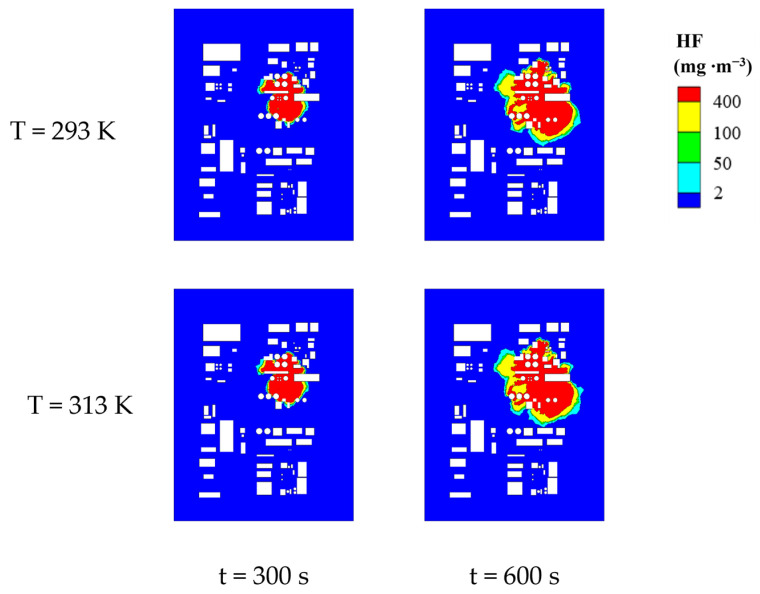
HF hazardous area distribution for complex scenarios (T = 293, 313 K, Y = 1.5 m).

**Figure 7 toxics-12-00184-f007:**
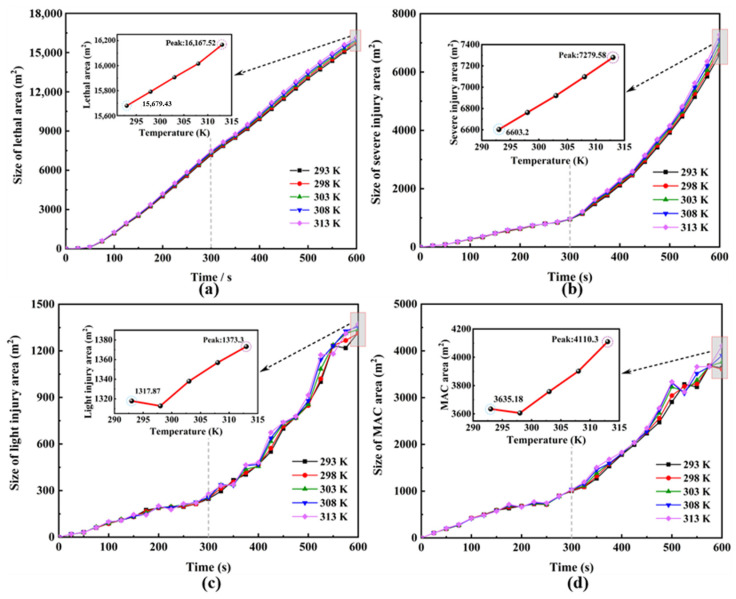
Change in hazardous area size for different ambient temperatures for complex scenarios (Y = 1.5 m); (**a**) Lethal area, (**b**) Severe injury area, (**c**) Light injury area, (**d**) MAC area.

**Figure 8 toxics-12-00184-f008:**
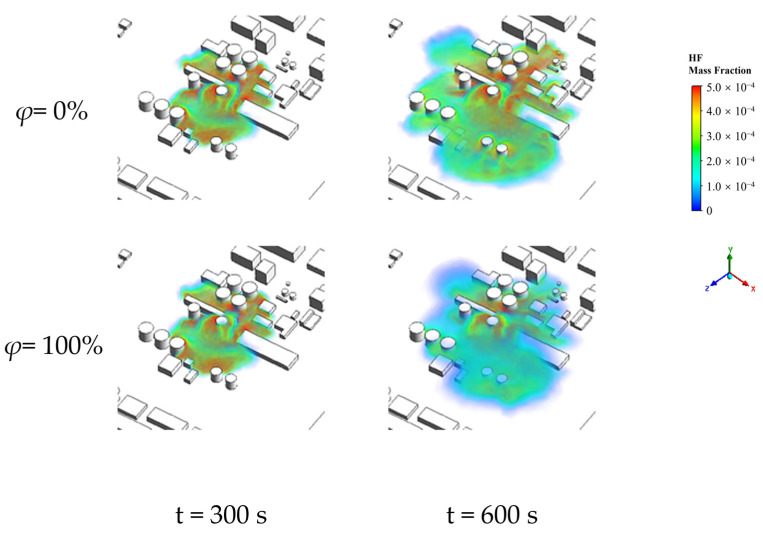
Complex scene HF 3D concentration distribution (*φ* = 0%, 100%).

**Figure 9 toxics-12-00184-f009:**
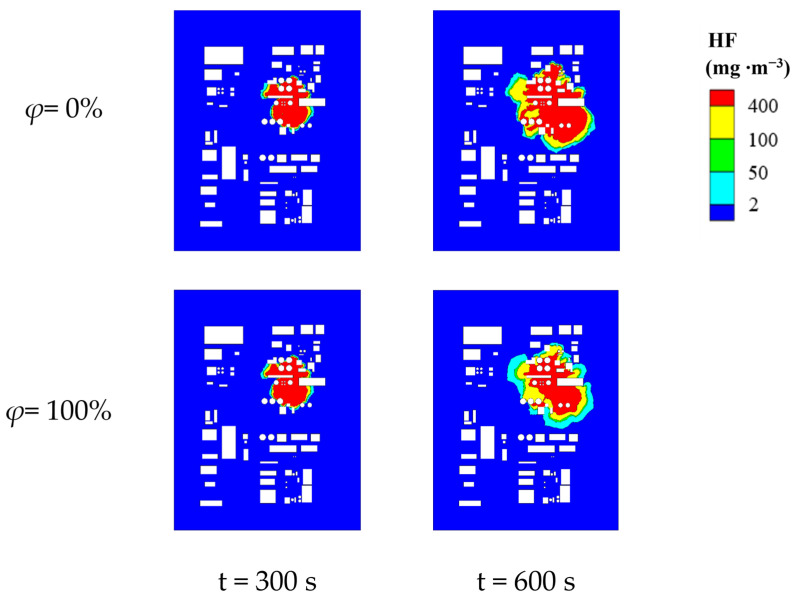
HF hazardous area distribution for complex scenarios (*φ* = 0%, 100%, Y = 1.5 m).

**Figure 10 toxics-12-00184-f010:**
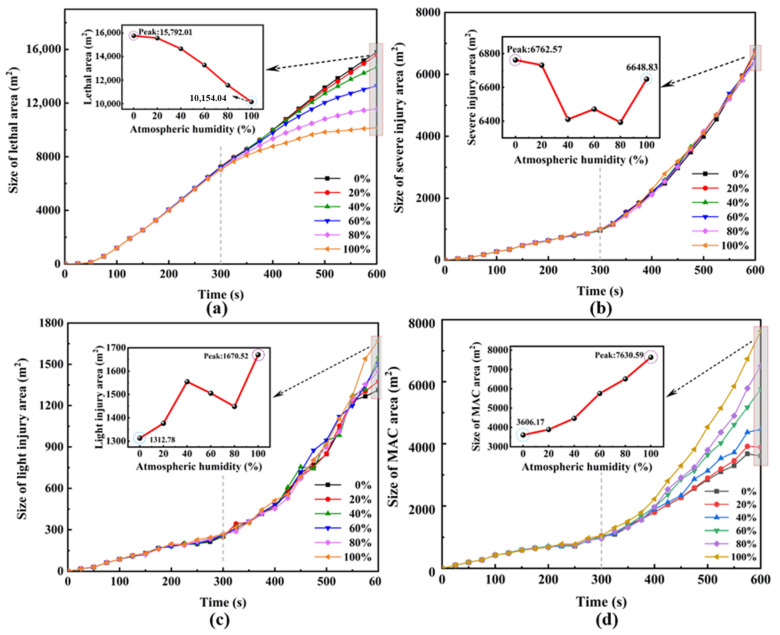
Change in hazardous area size at different relative humidities for complex scenarios (Y = 1.5 m); (**a**) Lethal area, (**b**) Severe injury area, (**c**) Light injury area, (**d**) MAC area.

**Table 1 toxics-12-00184-t001:** Conversion of real scenario wind levels to downscaled model wind speeds.

Level	Wind Speed (m/s)	Model Wind Speed (m/s)
0	0 ≤ *U_r_* ≤ 0.2	0 ≤ *U_m_* ≤ 0.043
1	0.2 < *U_r_* ≤ 1.5	0.043 < *U_m_* ≤ 0.32
2	1.5 < *U_r_* ≤ 3.3	0.32 < *U_m_* ≤ 0.71
3	3.3 < *U_r_* ≤ 5.4	0.71 < *U_m_* ≤ 1.2

**Table 2 toxics-12-00184-t002:** Information on the physical and chemical properties of hydrogen fluoride.

Name of Substance	HF	CO_2_
Boiling/K	292.66	216.15
Melting point/K	189.78	194.65
Relative Vapor Density(*ρ*_air_ = 1, 299.65 K)	1.7	1.5
Solubility in water	soluble	soluble
Vapor pressure (298.15 K)/mmHg	25	4.19 × 10^−5^
Stickiness(1–100 kPa, 273.15 K)/mPa·S	1.14 × 10^−2^	6.4 × 10^−2^
Toxicity level	4	/

**Table 3 toxics-12-00184-t003:** The basic information of CFD simulation operating conditions.

Tank Pressure (MPa)	Leakage Source Diameter (m)	HF Density (kg/m^3^)	Air Density (kg/m^3^)	Leakage Source Intensity (kg/s)
3	0.02	2.2	1.29	0.18

**Table 4 toxics-12-00184-t004:** Area statistics of various hazardous zones at 600 s of dispersion under different ambient temperatures.

Environmental Temperature /K	Size of Lethal Area/m^2^	Size of Severe Injury Area/m^2^	Size of Light Injury Area/m^2^	Size of MAC Area/m^2^	Size of Total Hazardous Area/m^2^
293	15,679.43	6603.2	1317.87	3635.18	27,235.68
298	15,792.01	6762.57	1312.78	3606.17	27,473.53
303	15,907	6920.88	1337.95	3757.85	27,923.68
308	16,016.22	7098.02	1356.97	3902.23	28,373.44
313	16,167.52	7279.58	1373.3	4110.3	28,930.70

**Table 5 toxics-12-00184-t005:** Area statistics of various hazardous areas at 600 s of dispersion under different atmospheric humidities.

Atmospheric Humidity/%	Size of Lethal Area/m^2^	Size of Severe Injury Area/m^2^	Size of Light Injury Area/m^2^	Size of MAC Area/m^2^	Size of Total Hazardous Area/m^2^
0	15,792.01	6762.57	1312.78	3606.17	27,473.53
20	15,590.19	6731.74	1376.47	3889.93	27,588.33
40	14,682.11	6409.93	1554.76	4467.84	27,114.64
60	13,288.35	6470.93	1505.1	5759.3	27,023.68
80	11,565.67	6392.83	1448.52	6510.89	25,917.91
100	10,154.04	6648.83	1670.52	7630.59	26,103.98

## Data Availability

The original data presented in the study are included in the article; further inquiries can be directed to the corresponding author.

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
