# Peer review of "The Effects of Ambient Temperature and Atmospheric Humidity on the Diffusion Dynamics of Hydrogen Fluoride Gas Leakage Based on the Computational Fluid Dynamics Method"

_toxics, 2024, doi:10.3390/toxics12030184_

Round 1

Reviewer 1 Report

Comments and Suggestions for Authors

In the paper, CFD was used to study the influence of temperature and humidity on the release and dispersion of hydrogen fluoride gas. It is a well-written paper, but the novelty of this paper is not clearly defined in the introduction. The novel elements of the research should be described in more detail in the introduction. In addition, several aspects should be corrected in the major revision of the manuscript:

1. Some additional comments on the 12 deaths in Weifang, Shandong, Rugao, Jiangsu, Ganzhou, Jiangxi, Guiyang, Guizhou, and Nantong, Jiangsu, which should be briefly described (see the Introduction).

2. What do you mean by "CFD simulation combined with Latin hypercube sampling statistical techniques to study the leakage and diffusion conditions of flammable gases under various circumstances"? (Lines 95-96)

3. How was the leakage rate experimental group selected with conditions of 2.5, 7.5, and 12.5 L/min?

3. The quality of Figure 1 is unacceptable. 

4. Fig.4 needs additional comments in the M&M part. 

5. Three-dimensional concentration distribution of RF for complex scenes should be explained in more detail in the Results and Discussion section. At present, it is described too laconically. 

6. 5. SI units should be used throughout the manuscript. For example, temperature is currently presented in both oC and K, but should be presented in K only.

7. Please avoid lump citations. For example, after a very general sentence stating that hydrogen fluoride gas could cause serious harm to human health and the environment, you cited 5 references. When you cite these references, you should mention in one or two words why you cite these papers. Without this explanation, such a citation is meaningless.

If released into the atmosphere, it can cause severe harm to human health, the en- 34 vironment, and infrastructure [1-5].

Reviewer 2 Report

Comments and Suggestions for Authors

The document titled "The Effects of Ambient Temperature and Atmospheric Humidity on the Diffusion Dynamics of HF Gas Leakage Based on the CFD Method" investigates how ambient temperature and atmospheric humidity impact the dispersion of hydrogen fluoride (HF) gas in a chemical industrial park using Computational Fluid Dynamics (CFD) method. It establishes a scaled-down experimental setup to validate numerical simulations under complex conditions, highlighting that atmospheric humidity significantly affects HF dispersion, unlike ambient temperature, which has a relatively minor impact.

If is possible try to apply the suggestions:

- Explore extreme environmental conditions beyond the studied ranges of 293 K to 313 K and 0% to 100% humidity to assess worse-case scenarios.

- Deepen the analysis of how specific humidity percentages influence the size and shape of hazard areas caused by HF dispersion.

- Pollution dispersion from a fire using a Gaussian plume model DOI 10.18280/ijsse.100401 

- Compare simulation results with data collected from real HF dispersion incidents to evaluate the model's prediction accuracy.

- Reduce the discrepancy between experimental and simulated values, potentially through boundary condition adjustments or simulation mesh refinement.

- Incorporate other environmental factors like wind speed, direction, and site topography to make the model more representative of real scenarios.

These recommendations aim to enhance the understanding of temperature and humidity effects on toxic gas dispersion and improve CFD simulation accuracy for safety planning and emergency responses.

Round 2

Reviewer 1 Report

Comments and Suggestions for Authors

The paper was corrected according my recommendtations.